# Clinical Outcomes of Complications Following Self-Expandable Metallic Stent Insertion for Benign Tracheobronchial Stenosis

**DOI:** 10.3390/medicina56080367

**Published:** 2020-07-22

**Authors:** Byeong-Ho Jeong, Jeffrey Ng, Suk Hyeon Jeong, Hojoong Kim

**Affiliations:** 1Division of Pulmonary and Critical Care Medicine, Department of Medicine, Samsung Medical Center, Sungkyunkwan University School of Medicine, Seoul 06351, Korea; myacousticlung@gmail.com (B.-H.J.); sukhyeon.jeong@samsung.com (S.H.J.); 2Division of Respiratory and Critical Care Medicine, University Medicine Cluster, National University Hospital, Singapore 119074, Singapore; jeffrey_sk_ng@nuhs.edu.sg; 3Yong Loo Lin School of Medicine, National University of Singapore, Singapore 119228, Singapore

**Keywords:** benign tracheobronchial stenosis, metallic stent, silicone stent

## Abstract

*Background and Objectives:* The use of metallic stents in benign TBS is controversial. Here, we report the clinical outcomes of patients who developed complications due to self-expandable metallic stent (SEMS) insertion for benign TBS. *Materials and Methods:* Our institution, which is the largest and most active referral hospital for airway stenosis in South Korea, only uses silicone stents. We conducted a retrospective review of 20 patients referred after the insertion of SEMS for benign TBS from 2006 to 2015. *Results:* All 20 patients underwent rigid bronchoscopy for SEMS removal due to airway obstruction from granulation tissue overgrowth. All but one (95%) experienced successful removal of the SEMS. During a median follow-up period of 40 months, a median of seven rigid bronchoscopies per patient was needed to maintain airway patency. Three (15%) patients suffered acute complications during SEMS removal (bleeding (10%) and fistula (5%)). All patients suffered chronic complications (granulation tissue (80%), stent migration (58%), mucostasis (55%), and restenosis (43%)). Eventually, 15 patients (75%) needed airway prostheses (silicone stent (75%) and tracheostomy (25%)). *Conclusion:* Our findings indicate that SEMS should be avoided until positive results are consistently reported by high-quality studies in patients with benign TBS.

## 1. Introduction

Currently, the majority of benign tracheobronchial stenosis (TBS) cases can be managed less invasively with interventional bronchoscopy rather than open surgery [1,2]. In 1987, Dumon developed a dedicated tracheobronchial silicone stent with multiple external studs that can be inserted and easily removed with rigid bronchoscopy [3]. Previous studies have shown favorable outcomes and easy customization with silicone stents in patients with benign TBS [4,5,6,7,8], thus, they have become the gold standard for benign TBS. However, the popularity of silicone stents has declined with the rise in the flexible bronchoscopy and recent expert opinion advocating the use of self-expandable metallic stents (SEMS) in selected cases, despite a lack of substantiating evidence [9,10].

The clinical use of SEMS and their commercial development is increasing. Through the 1990s, first-generation balloon-expandable metal stents progressed to second- and third-generation SEMS made from metal alloys such as nitinol, which confer shape memory properties [11,12]. The bio-mechanical properties of SEMS improved over the years; however, serious complications of SEMS began to arise due to stent-tissue interaction and cyclical compression force during normal respiration and coughing. Eventually, long-term maintenance of SEMS was found to cause granulation tissue overgrowth, airway trauma, erosion into adjacent structures, preclusion of surgery, difficult removal, and stent fracture. The Food and Drug Administration (FDA) has issued an advisory for consideration of other therapeutic options before insertion of metallic stents in benign TBS [9]. Subsequent to the FDA advisory, a tertiary referral center in the USA reported an increased number of cases referred for SEMS removal and in the associated high complication rates with significant costs and healthcare utilization rates, especially for SEMS placed in situ beyond 30 days [13].

However, SEMS have a major advantage due to their ease of deployment. In narrow stenotic lesions, the low-profile deployment catheter can be passed through the stenosis site and the SEMS can be deployed without having to risk perforation by positioning a rigid bronchoscope at the stenosis site. SEMSs also have a wider internal diameter and are better able to withstand angulation and buckling. This allows SEMS to have better conformation to complex airway stenotic anatomy and may help to form a better seal in anatomical dehiscence and fistula. These advantages have resulted in experts advocating the use of SEMS in benign TBS depending on operator expertise and stent availability [9].

Currently, there is no consensus on recommendations regarding the use of SEMS in benign TBS. Here, we report the clinical outcomes of patients who developed complications of SEMS insertion for benign TBS.

## 2. Methods

### 2.1. Patients

We retrospectively reviewed all patients who experienced complications after SEMS insertion for benign TBS between January 2006 and December 2015 at the Samsung Medical Center (a 1979-bed tertiary care referral hospital in Seoul, South Korea), which is the largest and most active referral hospital for airway stenosis in South Korea. During the study period, we did not use SEMS, even in cases of malignant TBS; therefore, all examined cases were referred from other hospitals.

Ethics approval was obtained on 8 August 2017 from the Institutional Review Board of Samsung Medical Center (IRB No. 2017-07-083) to review and publish data obtained from patient records. Informed consent was waived because patient information was anonymized and de-identified prior to analysis.

### 2.2. Airway Intervention Techniques

Airway anatomy was evaluated using chest computed tomography (CT), and when possible, flexible bronchoscopy. Airway intervention was performed according to the standard techniques described by Colt and Dumon [14] and Kim [15]. In short, patients were intubated with a rigid bronchoscope tube (Bryan Co., Woburn, MA, USA or Karl-Storz, Tuttlingen, Germany), and a flexible bronchoscope (BF 1T260 Olympus Corporation, Tokyo, Japan) was introduced through the rigid bronchoscope after induction of general anesthesia.

Depending on the characteristics of the airway complication related to the SEMS, various combinations of airway intervention techniques were used, including mechanical bougienage, ballooning, laser, and the insertion of silicone stents. Areas of fibrotic stenosis were gently dilated with a rigid bronchoscope (mechanical bougienage) or a controlled radial expansion balloon (Boston Scientific Corporation, Natick, MA, USA). A neodymium-doped yttrium aluminum garnet laser (LaserSonics, Milpitas, CA, USA) or a diode laser (Biolitec, Ceralas^®^, Jena, Germany) was used to cut the dense fibrotic bands. The SEMS was removed using optical rigid forceps. If SEMS removal with gentle traction was difficult, one side of the stent was longitudinally cut using optical scissors forceps, rolled, and then removed using optical rigid forceps. Finally, a silicone stent was inserted to maintain airway patency against fibrotic stenosis, overgrowth of granulation tissue, and malacia.

### 2.3. Airway Stents

The types of airway silicone stents used during the study included the Natural stent (M1S Co., Seoul, South Korea), Dumon stent (Novatech, La Ciotat, France), Y stent (M1S Co. or Novatech), and the Montgomery T-tube (Koken, Tokyo, Japan). The Natural stent is a silicone stent developed at Samsung Medical Center in 2002. Studies in a canine model of tracheal stenosis and clinical studies in patients with benign TBS have shown that the Natural stent is as effective and safe as the Dumon stent [16,17]. However, the production of Natural stents ceased due to commercial issues. Dumon stents have been commercially available for medical use since 2015.

### 2.4. Data Collection

We collected baseline data on patient demographics, etiology of TBS, stenosis site and severity, type of SEMS inserted, duration of SEMS in situ, and indication for removal. Performance status was evaluated using the American Society of Anesthesiologists (ASA) physical status classification: Class 1, a normally healthy patient; Class 2, a patient with mild systemic disease; Class 3, a patient with severe systemic disease that is not incapacitating; Class 4, a patient with an incapacitating systemic disease that is a constant threat to life [18]. The severity of airway stenosis was determined using the Myer-Cotton stenosis grading system: Grade I, ≤50% luminal stenosis; Grade II, 51–70% luminal stenosis; Grade III, 71–99% luminal stenosis; and Grade IV, no lumen [19]. Interventional data on whether patients underwent successful SEMS removal, additional stent insertion, laser therapy, balloon dilatation, or tracheostomy insertion was collected. Clinical outcomes were evaluated based on immediate symptom relief, spirometry data, chronic complications, and final status including persistent silicone stent placement, successful silicone stent removal, successful removal of SEMS without additional stenting, permanent tracheostomy, surgical management, and mortality.

### 2.5. Statistical Analysis

Continuous variables are presented as a median and interquartile range (IQR), and categorical variables are presented as a number (percentage). Spirometry data before and after the procedure were compared using the Wilcoxon signed rank test. Statistical differences were considered significant at *p* < 0.05. All analyses were performed using SPSS software (IBM SPSS Statistics ver. 25, Chicago, IL, USA).

## 3. Results

### 3.1. Baseline Characteristics

During the study period, 20 patients referred to the Samsung Medical Center for complications associated with SEMS insertion for benign TBS were enrolled. The median age was 40 years (IQR, 26–62 years) and 60% were female (Table 1). The etiology of benign TBS consisted of post-tuberculous tracheobronchial stenosis (PTBS; *n* = 7), post-intubation tracheal stenosis (PITS; *n* = 5), post-tracheostomy tracheal stenosis (PTTS; *n* = 5), post-operative tracheal stenosis (POTS; *n* = 1), traumatic bronchial rupture (*n* = 1), and tracheomalacia (*n* = 1). The patients underwent insertion of SEMS that had a median length of 55 mm (IQR, 45–78 mm). At the time of the first intervention at our institution, four (20%) patients suffered from respiratory failure and required intubation or tracheostomy. Nineteen covered and one uncovered SEMS were deployed for benign TBS involving the trachea (*n* = 14), left main bronchus (*n* = 5), and bronchus intermedius (*n* = 1). Most cases (85%) had ≥ 71% obstruction of the airway lumen (≥Grade III). The reasons for stent removal were granulation tissue overgrowth (100%), stent migration (20%), and stent fracture (5%), and removal took place a median of three months (IQR, 2–8 months) after SEMS insertion.

### 3.2. Treatment Modalities

During the median follow-up duration of 40 months (IQR, 19–88 months), a median of seven (IQR, 2–10) interventional bronchoscopies were needed to maintain airway patency (Table 2). After removal of the SEMS, most patients (95%) needed silicone stent insertion because of malacia and granulation tissue. Although all patients had stenosis at a single site (e.g., stenosis of the trachea only, left main only, or bronchus intermedius only) before insertion of SEMS, Y-stents were eventually required in six patients because of extensive damage to the bronchial cartilage around the main carina. In addition, balloon dilatation (*n* = 9), laser therapy (*n* = 5), or tracheostomy tube insertion (*n* = 5) were required during the treatment period.

### 3.3. Clinical Outcomes

Clinical outcomes are shown in Figure 1 and Table 3. The SEMS was successfully removed in 19 out of 20 patients. In the single failed case, broken parts of the uncovered SEMS could not be removed, and a silicone stent was inserted into the broken stent to maintain airway patency (Figure 2). Acute complications during SEMS removal occurred in three patients (15%) but were not difficult to manage. Most patients (95%) required silicone stent insertion to maintain airway patency after SEMS removal (Figure 3). All patients achieved immediate symptom relief after the first bronchoscopic intervention. During the treatment period, all patients experienced various chronic complications (granulation tissue overgrowth = 80%, stent migration = 58%, mucostasis = 55%, restenosis = 43%, malacia = 35%, and fistulas = 15%).

Eventually, 15 patients (75%) needed persistent stenting. They have all undergone periodic revision and silicone stent replacement over a median of 65 months (IQR, 35–106 months) for chronic complications. Permanent tracheostomies were performed in five of these 15 patients who needed frequent suctioning due to neurologic deficit (*n* = 3), subglottic stenosis (*n* = 1), and both (*n* = 1). Of these five patients with permanent tracheostomy, one died due to tracheostomy tube dislocation. Of the 19 patients who initially needed silicone stents, four were free of silicone stents after a median of 12 months (IQR, 7–22 months). No patient underwent surgical correction.

Spirometry tests were able to be performed in only eight patients before the first intervention at our institution and in all of the 15 patients without tracheostomy. Among the seven patients who had both test results, there were statistically significant increases in the ratio of forced expiratory volume in one second (FEV1) to forced vital capacity (from a median of 66 [IQR, 57–75] to a median of 83 [IQR, 68–94], *p* = 0.018) and the FEV1% predicted value (from a median of 83% [IQR, 68–94] to a median of 108% [IQR, 79–114], *p* = 0.042).

## 4. Discussion

Our institution is the most active hospital performing rigid bronchoscopy in South Korea. Recently, we treated 150 new patients with benign or malignant TBS and performed 400 rigid bronchoscopies in one year. Although we do not use SEMS in any type of TBS due to their disadvantages, some other institutions in South Korea use SEMS in patients with benign TBS as well as those with malignant TBS [20,21,22]. The use of SEMS in patients with benign TBS is controversial, therefore, we retrospectively reviewed the charts of 20 patients who experienced complications with SEMS. Although SEMS removal was successful in 19 (95%) patients, eventually 15 (75%) patients needed airway prostheses (5 patients with a tracheostomy and 15 patients with a silicone stent).

There are several reports regarding the clinical outcomes of patients with TBS undergoing SEMS insertion [13,23,24,25,26]. Some of these reports included a small number of malignant TBS cases [13,24,26], and one report preferentially attempted surgical correction [23]. The major indications for stent removal in these studies were granulation tissue overgrowth at the ends of the stents, purulent tracheobronchitis, stent fracture, fistulas, and stent migration. They reported stent removal success rates of 72–98% using mainly rigid bronchoscopy [13,24,25,26]. However, almost all patients suffered complications during and after stent removal. The main complications during stent removal were mucosal tearing and bleeding (5–25%), and a small number of tension pneumothoraces and fistulas were also reported [13,24,25,26]. During the follow-up period after stent removal, repeated interventions were needed to control the overgrowth of granulation tissue (~100%), and eventually 50–70% of patients underwent silicone stent insertion, tracheostomy, and/or surgical correction [13,24,25]. This is consistent with the results of our study in that SEMS can be safely removed using rigid bronchoscopy, but the incidence of chronic complications is high, even in skilled centers with multidisciplinary teams and expertise. In addition, higher complication rates appear to be associated with stents in place for > 30 days, benign TBS (vs. malignant TBS), and uncovered SEMS (vs. covered SEMS), so more attention should be paid to these cases [13,26].

Compared to our previous studies that reported the clinical course of benign TBS (PITS, PTTS, and PTBS) using silicone stent only [7,8,27], the median frequency of bronchoscopic intervention was higher (7 vs. 2–4), and various chronic complications were more common in this study. In the breakdown of chronic complications, rates of stent migration (58% vs. 35–52%) and restenosis (43% vs. 16–39%) were only slightly higher, while rates of granulation tissue overgrowth (80% vs. 40–65%) and mucostasis (55% vs. 18–28%) were appreciably higher in this population. Eventually, airway prostheses including silicone stents and tracheostomy tubes were more commonly required to maintain airway patency in this report (75% vs. 32–60%). These results may be attributable to the greater destructive effects of SEMS on airway mucosa as well as cartilage. The overall complication rate of SEMS is higher in benign TBS than in malignant TBS [28,29]. This may be because the stent is in situ for a longer period in patients with benign TBS, and the poor survival rate in patients with malignant TBS may not allow the development of chronic complications. Based on these results as well as the advantage of stent modification by cutting, silicone stents rather than SEMS are recommended in patients with benign TBS.

Despite concerns about the high complication rates associated with using SEMS in benign TBS, SEMS is often still preferred, mainly because of the ease of deployment, their self-expansile property, which allows better conformation to airway anatomy, and the larger internal diameter. In the midst of this controversy, there have been reports on third-generation SEMS in patients with benign TBS, and recently, several papers have shown good results. Dooms et al. retrospectively evaluated the clinical outcomes of several different third-generation SEMS in 17 patients with benign TBS [30]. They reported a short-term (within 12 weeks) complication rate of 75%, including migration (65%), fracture (15%), and wrinkling (10%) of stents and granulation tissue (10%). In contrast, Fortin et al. reported lower complication rates (migration 32.5%, granulation tissue 7.5%, subjective intolerance 5.0%, mucus plugging 2.5%, and laryngeal edema 2.5%) in 30 patients with benign TBS using a single type of third-generation SEMS (Silmet; Novatech, La Ciotat, France) [31]. The authors attributed the lower complication rate to more experience with a single type of stent over time and appropriate stent sizing. Contrary to the previous two reports in which fully covered SEMS were used for all patients, Xiong et al. reported the clinical outcomes of using uncovered and covered SEMS (Micro-Tech Co., Nanjing, China) in 72 and 59 patients with benign TBS, respectively [32]. Uncovered SEMS showed lower complication rates compared with covered SEMS (pain, 14% vs. 29%; major granulation tissue, 4% vs. 15%; recurrent stenosis, 10% vs. 29%, respectively). This is in contrast to previous studies that showed a higher incidence of complications with uncovered SEMS [26]. This difference may be due to accumulation of experience as well as advances in technology. Those authors indicated that uncovered SEMS have the theoretical advantage of neo-epithelialization within the stent, which is beneficial for mucociliary clearance. Although relatively positive results of SEMS have been reported in benign TBS in recent years, clinical translation of these results should be cautious because they are inconsistent and based on small retrospective studies.

Our study has several limitations. This study was retrospective and does not provide comparison data with silicone stents or malignant TBS. Selection bias could have affected our results as our data were collected from a single tertiary referral center. Our small patient sample (20 patients) is not representative of the whole population of patients who develop complications from metallic stenting for benign TBS. Covered SEMS were used in all except one of our study patients with an uncovered SEMS; hence, we are unable to provide outcome data for all types of SEMS that are currently available. In particular, the clinical outcome may vary depending on the specific properties or materials of the SEMS. Since there were no specific information on the SEMS, our results should be interpreted with caution.

## 5. Conclusions

Our analysis suggests that complications arising from SEMS insertion for benign TBS may result in increased morbidity, chronic complication rates, and the need for repeated bronchoscopic interventions. Although some articles have reported good results with SEMS, we suggest that the use of SEMS be avoided in patients with benign TBS until positive results are consistently reported by high-quality studies.

## Figures and Tables

**Figure 1 medicina-56-00367-f001:**
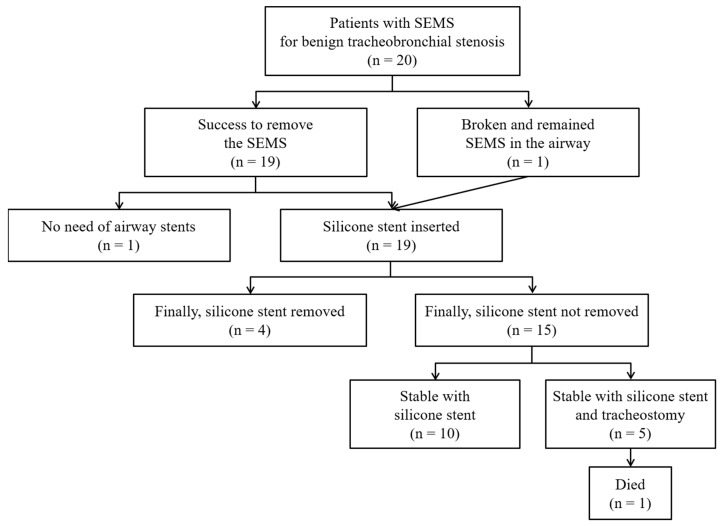
Clinical course of patients with a self-expandable metallic stent (SEMS) for benign tracheobronchial stenosis.

**Figure 2 medicina-56-00367-f002:**
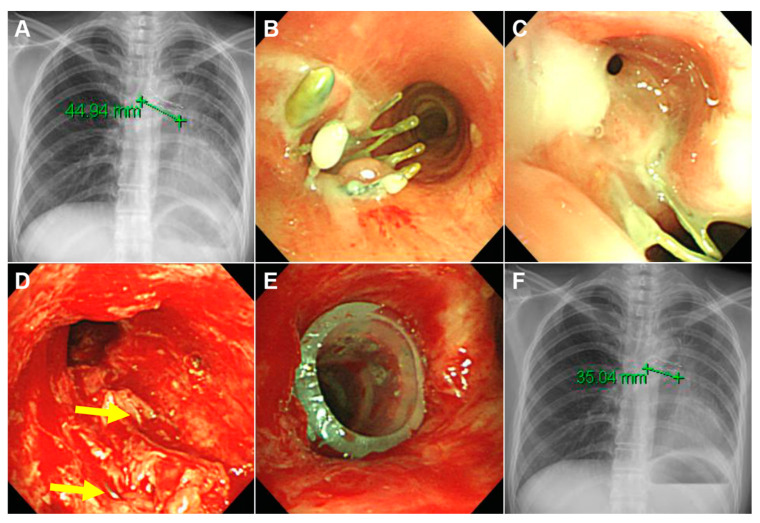
A case in which the SEMS could not be completely removed. A 47-year-old female complained of progressive dyspnea. She was treated for endobronchial tuberculosis and underwent insertion of an uncovered SEMS in the left main bronchus 17 years prior. (**A**) Chest radiography showed a 45-mm SEMS in the left main bronchus. Her left upper lobe was already completely destroyed on chest computed tomography. (**B**,**C**) Her left main bronchus was almost totally obstructed by granulation tissue overgrowth. (**D**) We tried to remove the SEMS, but most of the stent was embedded (yellow arrows) in the deep layer of the bronchus. (**E**) After mechanical dilatation, a Natural stent (outer diameter = 10 mm, length = 45 mm) was inserted into the left main bronchus. (**F**) After the procedure, the remaining SEMS (length = 35 mm) was visible on chest radiography.

**Figure 3 medicina-56-00367-f003:**
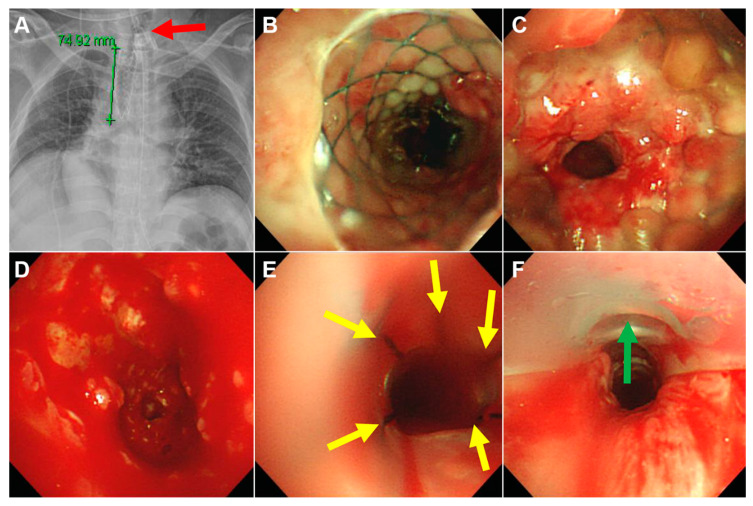
A case in which the SEMS was successfully removed but a silicone stent was needed to maintain airway patency. A 35-year-old male who had undergone tracheostomy nine years prior due to a traffic accident suffered from progressive dyspnea after tracheostomy closure and subsequently underwent SEMS insertion. However, he experienced progressive respiratory distress again two months after stent insertion. The patient was referred to our institution with a tracheostomy. (**A**) Chest radiography showed a 75-mm SEMS in the trachea and a tracheostomy tube (red arrow). (**B**) There was no stenosis at the upper end of the SEMS. (**C**) However, there was Grade III stenosis at the lower end of the SEMS. (**D**) The SEMS was rolled up and pulled out using optical rigid forceps. Severe stenosis and granulation tissue remained in the trachea. (**E**) We inserted two Natural stents with an outer diameter of 12 mm and a length of 80 mm (sewn 50 mm and 30 mm stents using black silk (yellow arrows)). (**F**) Then, we inserted a tracheal stoma retainer through the tracheostomy site (green arrow).

**Table 1 medicina-56-00367-t001:** Clinical characteristics at the time of the first intervention at our institution.

Variables	*n* = 20
Age, year	40 (26–62)
Sex, female	12 (60)
Comorbidities	
No	6 (30)
Old tuberculosis	7 (35)
Cerebrovascular disease	4 (20)
Diabetes	3 (15)
Lobectomy status *	2 (10)
Leukemia	1 (5)
Etiology of benign tracheobronchial stenosis	
Post-tuberculous tracheobronchial stenosis	7 (35)
Post-intubation tracheal stenosis	5 (25)
Post-tracheostomy tracheal stenosis	5 (25)
Others ^†^	3 (15)
ASA physical status ^‡^	
Class 1	2 (10)
Class 2	12 (60)
Class 3	4 (20)
Class 4	2 (10)
Intubation or tracheostomy due to respiratory failure before intervention	4 (20)
Stenosis site	
Trachea	14 (70)
Left main bronchus	5 (25)
Bronchus intermedius	1 (5)
Severity of stenosis (Myer and Cotton Grade) ^§^	
I	1 (5)
II	2 (10)
III	14 (70)
IV	3 (15)
Spirometry (*n* = 8)	
FEV_1_/FVC	69 (58–75)
FEV_1_, L	1.95 (1.68–2.10)
FEV_1_, % predicted	78 (67–88)
FVC, L	2.79 (2.49–3.00)
FVC, % predicted	86 (72–103)
Type of SEMS	
Covered SEMS	19 (95)
Uncovered SEMS	1 (5)
Length of SEMS, mm	55 (45–78)
Duration of SEMS in place, months	3 (2–8)
Reason for removal of SEMS	
Granulation tissue overgrowth	20 (100)
Stent migration	4 (20)
Stent fracture	1 (5)

Data are presented as *n* (%) or the median (interquartile range). ASA = American Society of Anesthesiologists; FEV1 = forced expiratory volume in 1 sec; FVC = forced vital capacity; SEMS = self-expandable metallic stent. ***** Diagnostic lobectomy of left upper lobe for indeterminant nodule (*n* = 1), and therapeutic lobectomy of left lower lobe for bronchiectasis (*n* = 1). ^†^ Post-operative tracheal stenosis (*n* = 1), traumatic bronchial rupture (*n* = 1), and tracheomalacia (*n* = 1). ^‡^ Class 1, a normally healthy patient; Class 2, a patient with mild systemic disease; Class 3, a patient with severe systemic disease that is not incapacitating; Class 4, a patient with an incapacitating systemic disease that is a constant threat to life. ^§^ Categorization is based on the percentage of reduction in cross-sectional area. Grade I, ≤50% luminal stenosis; Grade II, 51–70% luminal stenosis; Grade III, 71–99% luminal stenosis; and Grade IV, no lumen.

**Table 2 medicina-56-00367-t002:** Treatment modalities during follow-up.

Variables	*n* = 20
Duration of follow-up after the first intervention at our hospital, months	40 (19–88)
Number of interventional bronchoscopies	7 (2–10)
Treatment modalities *	
Successful removal of SEMS	19 (95)
Silicone stent insertion	19 (95)
Dumon or Natural stent	17/19 (89)
Y-stent	6/19 (32)
Montgomery T-tube	2/19 (11)
Ballooning	9 (45)
Laser therapy	5 (25)
Tracheostomy tube insertion	5 (25)

Data are presented as *n* (%) or the median (interquartile range). SEMS = self-expandable metallic stent. ***** Patients could undergo more than one procedure.

**Table 3 medicina-56-00367-t003:** Clinical outcomes.

Variables	*n* = 20
Immediate symptom relief	20 (100)
Acute complications during the SEMS removal	3 (15)
Mucosal tear and bleeding	2 (10) *
Tracheo-esophageal fistula	1 (5) ^†^
Chronic complications during the follow-up	20 (100)
Granulation tissue overgrowth	16 (80)
Stent migration	11/19 (58) ^‡^
Mucostasis	11 (55)
Malacia	7 (35)
Tracheoesophageal fistula	3 (15)
Restenosis after stent removal	3/7 (43) ^§^
Spirometry in the last state (*n* = 15)	
FEV_1_/FVC	71 (65–84)
FEV_1_, L	2.50 (1.89–2.86)
FEV_1_, % predicted	88 (79–108)
FVC, L	3.26 (2.61–3.85)
FVC, % predicted	92 (77–104)
Final outcomes	
Persistent silicone stent placement	15 (75)
Duration of stent placement, months	65 (35–106)
Successful silicone stent removal	4 (20)
Duration of stent placement, months	12 (7–22)
Duration of follow-up after stent removal, months	6 (4–26)
Successful removal of SEMS without additional stenting	1 (5) ^¶^
Permanent tracheostomy	5 (25)
Surgical management	0 (0)
Mortality	1 (5) ^ǁ^

Data are presented as *n* (%) or the median (interquartile range). SEMS = self-expandable metallic stent; FEV1 = forced expiratory volume in 1 sec; FVC = forced vital capacity. ***** These cases needed admission to the intensive care unit. ^†^ The fistula spontaneously healed after two weeks of fasting. ^‡^ Excluded one patient who did not require additional stenting after SEMS removal. ^§^ Seven patients who underwent stent removal at least once are the denominator. ^¶^ This patient underwent rigid bronchoscopy with ballooning and laser treatment for bronchial stricture eight months after SEMS removal. ^ǁ^ This patient died after tracheostomy tube dislodgement 11 months after SEMS removal.

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
