# Peer review of "Clinical Outcomes of Complications Following Self-Expandable Metallic Stent Insertion for Benign Tracheobronchial Stenosis"

_medicina, 2020, doi:10.3390/medicina56080367_

Round 1

Reviewer 1 Report

This is a very nice retrospective review of complications related to metallic stents for benign tracheobronchial stenosis. It is interesting because it presents a fair number of patients treated during a period of 10 years and followed for 40 months.

I only have minor suggestions, particullarly regarding Table 1, 2 and 3. I find presentation of data confusing, mainly related to the way information is depicted (centered) and also, categorization of information. The second column shows both percentages and ranges without clarifying which is which. Finality of tables is to show a summery at a glance, and I had to take time to understand all tables. I would suggest to modify them, make them simpler and easier to undestan, and difine either percentages or ranges in the correspondent column. 

Conclusions are very appropriate, to avoid SEMS in benign conditions of the airway.

Author Response

C1. This is a very nice retrospective review of complications related to metallic stents for benign tracheobronchial stenosis. It is interesting because it presents a fair number of patients treated during a period of 10 years and followed for 40 months.

R1. Thank you for encouraging our results

C2. I only have minor suggestions, particullarly regarding Table 1, 2 and 3. I find presentation of data confusing, mainly related to the way information is depicted (centered) and also, categorization of information. The second column shows both percentages and ranges without clarifying which is which. Finality of tables is to show a summery at a glance, and I had to take time to understand all tables. I would suggest to modify them, make them simpler and easier to undestan, and difine either percentages or ranges in the correspondent column.

R2. I totally understand your concerns. The current format of these tables (center-alignment) was modified by the Medicina Journal. I looked up other articles of the Medicina Journal, and found that they were also centrally aligned in the Tables. I also want to use left-alignment in the Tables to avoid confusion, but the Journal’s official format seems to be center-alignment.

C3. Conclusions are very appropriate, to avoid SEMS in benign conditions of the airway.

R3. Thank you again for supporting our results.

Reviewer 2 Report

General (major)

  • The “SEMS” is just one type of many available metallic stents, it not appropriately represented all the “metallic stents”. This manuscript only studies the “SEMS”. Therefore, there are many “metallic stents” should be replaced by the “SEMS”, include in the “Title”.
  • Many factors, such as stent stiffness, dynamic, mucosal tissue sensitivity, etc., that may influent the clinical outcomes. Different materials of the SEMS also have different tissue reactions and outcomes. There may enrich with what kind of material made for this “SEMS”.

Minor

In “Introduction”

  • Should more focus and concentrated on the “SEMS” instead of the “silicon stent”.

In “Results”

  • All these 3 Tables are not well arranged (variable columns) as usually.
  • In the last sentence of “…which occurred a median….”, may change to “…which noted a median…”.

Author Response

General (major)

C1. The “SEMS” is just one type of many available metallic stents, it not appropriately represented all the “metallic stents”. This manuscript only studies the “SEMS”. Therefore, there are many “metallic stents” should be replaced by the “SEMS”, include in the “Title”.

R1. I understand your concerns and totally agree with your opinion. To clarity the meaning, the term ‘metallic stents’ used in many places has been changed to ‘SEMS’, including the text as well as the Title, Tables, and Figures.

C2. Many factors, such as stent stiffness, dynamic, mucosal tissue sensitivity, etc., that may influent the clinical outcomes. Different materials of the SEMS also have different tissue reactions and outcomes. There may enrich with what kind of material made for this “SEMS”.

R2. You are right. All 20 cases in this report were treated with SEMS at other hospitals as described in the Method section, unfortunately, we had very limited information about the SEMS. So, we have already described these limitations in the Discussion section. However, as your opinion, we felt that there was a lack of description on the detailed information of the SEMS, we have further described the limitations of this report in the Discussion section as follow (Line 283-285): “In particular, the clinical outcome may vary depending on the specific properties or materials of the SEMS. Since there were no specific information on the SEMS, caution should be taken in interpreting our results.”

Minor

C3. In “Introduction”

Should more focus and concentrated on the “SEMS” instead of the “silicon stent”.

R3. Thank you for improving the quality of our work. We modified the Introduction section (Line 33-34, 40-45, 49-57) as your recommend.

C4. In “Results”

All these 3 Tables are not well arranged (variable columns) as usually.

R4. I totally understand your concerns. The current format of these tables (center-alignment) was modified by the Medicina Journal. I looked up other articles of the Medicina Journal, and found that they were also centrally aligned in the Tables. I also want to use left-alignment in the Tables to avoid confusion, but the Journal’s official format seems to be center-alignment.

C5. In the last sentence of “…which occurred a median….”, may change to “…which noted a median…”.

R5. Thank you for giving me a better expression. We changed the sentence as your recommend in Line 129.